# Exploring commercial GNSS RO products for Planetary Boundary Layer studies in the Arctic

Manisha Ganeshan[1,2], Dong L. Wu[1], Joseph A. Santanello[3], Jie Gong[1], Chi Ao[4], Panagiotis Vergados[4] and Kevin J. Nelson[4]

[1]Climate and Radiation Laboratory, NASA Goddard Space Flight Center, Greenbelt, 20771, USA
[2]Morgan State University, Baltimore, 21251, USA
[3]Hydrological Sciences Laboratory, NASA Goddard Space Flight Center, Greenbelt, 20771, USA
[4]NASA Jet Propulsion Laboratory, California Institute of Technology, Pasadena, 91109, USA

*Correspondence to*: Manisha Ganeshan (manisha.ganeshan@nasa.gov)

**Abstract.**

Commercial Radio Occultation (RO) satellites that utilize the Global Navigation Satellite System (GNSS) signals are emerging as key tools for observating the polar regions, which are not covered by the second-generation Constellation Observing System for Meteorology, Ionosphere, and Climate (COSMIC-2) mission. This study evaluates the value of commercial RO measurements, specifically Spire and GeoOptics, for planetary boundary layer (PBL) investigations in the Arctic, a region where favourable lower atmospheric penetration of GNSS RO is vital for observing the persistently shallow PBL. The lower tropospheric penetration capability of both Spire and GeoOptics over the Arctic Ocean, with nearly 80% observations reaching an altitude of 500 meters above mean sea level, is comparable to other RO missions such as the current Meteorological Operational Satellite (MetOp) and the discontinued COSMIC-1 missions. A seasonal cycle in RO penetration probability, with minimum occurring during the Arctic warm season, was observed in most RO datasets, except NASA-purchased Spire data. Monthly mean Arctic PBL height (PBLH) derived from Spire and GeoOptics compares well with MetOp observations and the reanalysis from Modern-Era Retrospective analysis for Research and Applications version 2 (MERRA-2). A minimum penetration threshold of 500 meters generally suffices for determining Arctic PBLH, although 300 meters threshold improves performance of NASA-purchased Spire data. Arctic PBLH representation is influenced less by the number of observations or instrument type and more by the algorithms used for bending angle and refractivity retrievals. These findings underscore the importance of processing algorithms in achieving accurate lower atmospheric soundings and Arctic PBLH representation.

## 1 Introduction

The planetary boundary layer (PBL) is a target observable of broad importance to the Earth Science community. The Global Navigation Satellite System (GNSS) Radio Occultation (RO) has been shown to be a good candidate for observing the

PBL height (PBLH) across various spatiotemporal scales (Ao et al., 2012; Basha and Ratnam, 2009; Ding et al., 2021; Kalmus et al., 2022; Nelson et al., 2021; Winning et al., 2017) as recommended by the National Academies of Science Decadal Survey for Earth Science and Applications from Space report (NASEM, 2018; Teixeira et al., 2021). Today, advancing PBL science is inherently reliant on high resolution observations with high-frequency sampling that can chiefly be afforded by a single remote sensing instrument/combination of instruments from space. In this regard, GNSS RO is a vital measurement technique, due to its superior vertical resolution (< 100 m) and viewing geometry compared to most other nadir-viewing space-based instrument technologies, allowing penetration down to 100 meters above surface. High vertical resolution measurements and deep penetration of observations into the lower atmosphere are deemed vital. This capability is particularly valuable for polar regions where persistent surface-based temperature inversions create shallow PBLs that are difficult to observe using other remote sensing methods.

## 1.1 Importance of GNSS RO for Arctic PBL studies: Why commercial data?

The study of the Arctic Ocean PBL can greatly benefit from GNSS RO observations due to their ability to: (a) operate under all weather conditions, (b) penetrate persistent cloud cover, (c) perform effectively over flat surfaces like sea ice and open ocean, and (d) contribute to a long-term climate data record enhanced by commercial satellite coverage. With the decommissioning of the Constellation Observing System for Meteorology Ionosphere and Climate (COSMIC-1) in 2019 and the limited (45°N to 45°S) latitude range of COSMIC-2, commercial satellites provide critical high-latitude observations. However, to fully leverage their potential, it is essential to evaluate their lower atmospheric sounding capabilities and ensure their compatibility with existing climate data records. The refractivity gradient method, commonly used for determining the PBL height (Ao et al., 2012; Nelson et al., 2021; Qiu et al., 2023; Seidel et al., 2010, 2012), is found to be sensitive to the penetration capability of RO profiles in the Arctic (Ganeshan and Wu, 2015). From the analysis of 8 years of COSMIC-1 data, it was found that availability of RO profiles over the Arctic Ocean reduced significantly at tangent heights below 1km, which introduces a sensitivity of the retrieved PBL height to the choice of the cutoff altitude, or minimum RO penetration depth, used for profile selection. It was noted that only the absolute PBLH values were sensitive to the choice of cutoff altitude, whereas the spatial and seasonal variability remained largely unaffected (Ganeshan and Wu, 2015). Thus, it is worthwhile exploring the lower atmospheric penetration capability of commercial RO products and their representation of the Arctic PBLH compared to past and current existing operational GNSS RO products.

## 1.2 A background of GNSS RO neutral atmosphere technique

In the GNSS RO retrieval technique, the neutral atmosphere is considered as the atmospheric path consisting of the troposphere and stratosphere (up to 60 km) which is refractive and electrically neutral, unlike the mesosphere and ionosphere-thermosphere regions. The neutral atmosphere has both dry and wet components that contribute to refraction, with the wet component becoming more important closer to the surface due to increased concentrations of water vapor. Not all RO profiles reach the surface, and in fact, there can be an exponential drop in the fraction of available RO observations (i.e., penetration

probability) as we go towards the surface as shown by Ganeshan and Wu (2015). This change in penetration probability is primarily due to the decrease in the signal-to-noise ratio (SNR) caused by atmospheric defocusing (Wu et al., 2022). However, factors such as instrument design, neutral atmosphere excess phase computation method, and choice of bending angle retrieval algorithm can also affect the penetration probability profile for a given atmospheric path (Vannah et al., 2025).

A thorough understanding of factors affecting RO penetration is desirable to help minimize sampling bias as well as to ensure data continuity and consistency in climate records. However, this is difficult to achieve, given the existence of a large number of GNSS RO missions and different versions of products from a single mission that are periodically re-processed to remain up to date with advances in software and processing algorithms (Vannah et al., 2025). This study aims to provide a comparison of the penetration capability of new commercial GNSS RO data products against other existing products in the

Arctic as the first step towards establishing a climate ready, long-term continuous, dataset that can be used for Arctic PBL investigations.

## 2 Data and Methodology

### 2.1 Datasets

The value of commercial GNSS RO products for PBL studies in the Arctic Ocean (north of 60°N, excluding land

areas) is assessed by comparing with established RO mission products, such as, COSMIC-1 and the Meteorological Operational satellite programme (MetOp), as well as reanalyses data. The commercial GNSS RO data evaluated in this study are purchased by NASA through the Commercial SmallSat Data Acquisition (CSDA) program. As of this study, the only approved vendors with radio occultation data in NASA's CSDA program are Spire and GeoOptics, though data from other vendors is under evaluation for inclusion in the CSDA archives (NASA CSDA, 2025). In addition, commercial data for near-

real-time operations purchased by NOAA and processed by UCAR are also analyzed for overlapping periods.

Table 1 summarizes the RO datasets used, including data periods, processing centers, and average monthly profiles over the Arctic Ocean.

### 2.1.1 Commercial RO Data

NASA-purchased Spire data are processed by the vendor, and provided at a similar vertical grid and resolution as

other GNSS RO missions (such as COSMIC, COSMIC-2, and MetOp) where the lowest level of valid observations differs from profile to profile, because the penetration depth achieved by each RO is unique, depending primarily on the SNR. GeoOptics data purchased under the CSDA program and provided by the vendor, on the other hand, are on a uniform 100 m vertical grid, along with a quality flag that is used to determine the lowest penetration level. GeoOptics uses the phase matching methodology in RO processing (Jensen et al., 2004), a wave optics technique designed to extract the full information from the

received wave field. The quality flag is applied in two ways: (i) blanket criteria that checks the range of the amplitude of computed phase matching integral and cumulative number of phase jumps within the upper neutral atmosphere (between 8 to 40 km),  rejecting profiles if the previous checks failed, and (ii) individual criteria that flag each level as "good" or "bad" based

on the presence or absence of sharp features (e.g., moisture and temperature gradients) that can cause significant deviation of the bending angle relative to a smoothed background bending angle profile. In this study, only profiles satisfying the blanket criteria are considered. Additionally, for each profile, the minimum penetration depth is ascertained by the lowest above-surface level with a "good" quality flag. It is important to note that if a "sharp" PBL inversion layer with poor quality control (QC) flag exists above the minimum penetration depth, that profile is not discarded.

The NOAA Spire and GeoOptics data purchased for near-real-time operations are downloaded from the University Corporation for Atmospheric Research (UCAR, UCAR COSMIC Program, 2025 a,b) website. NOAA purchases Level 1b (L1B) data from both vendors and the Level 2 (L2) neutral atmosphere products are retrieved from in-house excess phase computations carried out by UCAR in near-real-time. In the case of GeoOptics, the overlap between available NOAA and NASA purchased data is during the month of April 2021, and for Spire, the month of October 2021 is chosen to compare overlapping data. All subsequent references to "Spire" and "GeoOptics" in this paper imply NASA-purchased commercial RO data unless explicitly specified to be NOAA-purchased datasets.

### 2.1.2 Other RO Datasets

MetOp data from the Radio Occultation Meteorology Satellite Applications Facility (ROM SAF) and COSMIC-1 data from UCAR (2013 and 2021 re-processed versions) are used in this study for comparative analysis. A major focus will be year-long comparisons between NASA Spire, NASA GeoOptics, and the re-processed MetOp data from ROM SAF. The MetOp data are part of the Interim Climate Data Record (ICDR) ROM SAF product which was developed in 2017 (ROM SAF, 2019). Although the MetOP near-real-time (NRT) product from ROM SAF has more advanced processing setup with improved lower tropospheric penetration, the goal is to compare with a consistent climate record to avoid ambiguities resulting from frequent software updates. Therefore, the ICDR data are used in this study. Some differences are observed between the rising and setting occultations of MetOp data owing to the use of raw sampling tracking which is not considered full "open loop" tracking. In this study, we only consider setting occultations from MetOp which are known to have slightly better SNR and an overall deeper penetration (Innerkofler et al. 2023). Additionally, re-processed data from COSMIC-1 available from the University Corporation for Atmospheric Research (UCAR Data Release, 2022) are used to compare RO penetration statistics over the Arctic. COSMIC-1 data ceased to be produced beyond 2019, thereby limiting their use for this comparative analysis which is mainly focused on the year 2020. For this study, they serve to provide a stable climatological record of RO penetration statistics over the Arctic Ocean against which characteristics of newer datasets can be compared. Two versions of UCAR reprocessed COSMIC-1 data (from the year 2013 and the year 2021) are obtained for the period ranging from 2007 to 2013 and from 2007 to 2017, respectively. Table 1 lists and describes all RO datasets used in this study, including the center where the L2 data are processed.

**Table 1** List of RO satellite products used in this study including the Level 2 data processing center, data version, and processing mode, along with the chosen study period and average total monthly RO profile count available over the Arctic Ocean during the study period.

| Satellite Product | Processing Center | Data Range | Average Monthly Profile Count | Data Version | Processing Mode |
|---|---|---|---|---|---|
| MetOp ICDR | ROM SAF | 2020, Apr 2021, Oct 2021 | 1974 | ICDR | Re-Processed |
| COSMIC 2013 | UCAR | 2007-2013 (only Apr and Oct) | 3503 | 2013.3520 | Re-Processed |
| COSMIC 2021 | UCAR | 2007-2017 (only Apr and Oct) | 2904 | 2021.0390 | Re-Processed |
| NASA Spire | Spire | 2020, Oct 2021, Feb 2024 | 17207 | Version 06 | Vendor-Provided |
| NOAA Spire | UCAR | Oct 2021, Feb 2024 | 6223 | - | Near-Real-Time |
| NASA GeoOptics | GeoOptics | 2020, Apr 2021 | 754 | Version 01 | Vendor-Provided |
| NOAA GeoOptics | UCAR | Apr 2021 | 3250 | - | Near-Real-Time |

*2.1.3 Reanalysis Data*

The Modern-Era Retrospective analysis for Research and Applications version 2 (MERRA-2) reanalysis product (Gelaro et al., 2017) is used to obtain the monthly mean PBLH and the monthly mean sea ice fraction over the Arctic Ocean, to compare against the PBLH derived from GNSS RO datasets. The GEOS atmospheric model, used in MERRA-2, has a horizontal resolution of approximately ~0.5 degrees. The model vertical grid is based on a terrain-following sigma coordinate system, with the first model level over the Arctic Ocean typically being around 50 meters above surface and the vertical grid spacing around 100 meters within the lowest five model levels.

In MERRA-2, the PBLH is defined as the model level where the eddy heat diffusivity coefficient ($K_H$) value falls below 2 $m^2$ $s^{-1}$ threshold (McGrath-Spangler et al., 2015). The GEOS atmospheric model used in MERRA-2 uses the non-local Lock scheme (Lock et al. 2000), in conjunction with the first-order local turbulence closure scheme Louis scheme (Louis et al. 1982). The Lock scheme is used to parameterize non-local mixing in unstable layers simulating the effects of surface heating and boundary layer cloud-top cooling, including entrainment, whereas the Louis scheme treats both stable and unstable boundary layers. The scheme estimates heat and momentum diffusivity coefficients based on bulk Richardson number and the

turbulent length scale, with provision for dependency on the PBLH from the previous time step in case of unstable layers (Ganeshan and Yang, 2019). During persistent stable conditions, such as are commonly observed over the frozen Arctic Ocean, the turbulent length scales are expectedly small, implying that the model diffusivity coefficients are largely based on the bulk Richardson number. Thus, MERRA-2 PBLH over Arctic is expected to be sensitive to wind and temperature gradients (used for computing the bulk Richardson number), making it comparable to the refractivity-based RO PBLH which responds primarily to the temperature inversion (Ganeshan and Wu, 2015).

## 2.2 PBLH Determination from RO datasets

GNSS RO-derived PBLH is calculated using a gradient method that identifies the first refractivity gradient minimum exceeding -40 N-units km$^{-1}$. A standard cutoff altitude of 500 meters is applied to ensure sufficient penetration depth, although sensitivity to lower thresholds (e.g., 300 meters) is also assessed.

The PBLH is derived from the GNSS RO refractivity profile using the bottom-up search approach described in Ganeshan and Wu (2015), identifying the first minima of the refractivity gradient to exceed -40 N-units km$^{-1}$ and assigning the corresponding altitude as the PBLH. This approach is specifically useful for deriving the height of the PBL inversion over the Arctic during cold season months. A cutoff altitude threshold (which is a required minimum penetration threshold), typically set to 500 m (Ao et al., 2012, Guo et al., 2011, Ganeshan and Wu, 2015, Nelson et al., 2021), is applied so that only RO profiles that reach this altitude or lower are included. Ganeshan and Wu (2015) showed that even though the magnitude of the retrieved PBLH over Arctic is sensitive to the cutoff altitude, its spatiotemporal variability remained unaffected by the choice of this threshold. In this study, sensitivity of commercial RO products to the choice of cutoff altitude threshold will be additionally explored. All GNSS RO-derived monthly mean penetration probability and monthly PBL height characteristics are interpolated onto a 2° latitude x 10° longitude grid, as in Ganeshan and Wu (2015). A distance-weighted averaging method is used for interpolation by considering observations falling within a circle of 5° around each grid point. The MERRA-2 variables are similarly interpolated onto the 2°x10° horizontal grid for ease of comparison.

## 3 Results and Discussion

### 3.1 Sensitivity of RO penetration loss to bending angle processing method

GNSS RO bending angle and refractivity profile observations are characterized by a loss of signal (decrease in SNR) as they approach the surface due to atmospheric defocusing (Wu et al., 2022). However, the rate of penetration loss is expectedly different for various RO missions, due to diversity in the design of GNSS receivers and their SNR capabilities. Penetration loss can also be different for measurements from the same instrument due to the viewing geometry, as rising occultations can be more difficult to track (Innerkofler et al. 2023), as well as due to inherent disparity in excess phase computations and bending angle retrieval algorithms (Vannah et al., 2025). For example, older versions of the same product,

such as the UCAR COSMIC 2013 version, can differ significantly from newer reprocessed versions (e.g., COSMIC 2021 version), due to advances in excess phase computations, retrieval software, GNSS orbits, clock, and earth orientation products (UCAR Data Release, 2022).

GNSS RO penetration loss varies by mission and processing methodology. Figure 1(a) compares the rate of RO penetration loss over the Arctic Ocean for different GNSS RO missions (COSMIC, MetOp, Spire) as well as for different products from the same mission (COSMIC 2013 vs. COSMIC 2021; Spire NASA vs. Spire NOAA). Clearly, the penetration loss is much less significant for the newer version of COSMIC-1 data compared to the older version. The penetration probability is more or less similar for Spire, MetOp, and COSMIC 2021 products, with differences generally being confined to the lowest 1 km. Spire NASA and MetOp penetration probabilities behave similarly. On the other hand, Spire NOAA data are similar to the re-processed COSMIC 2021 product despite differences in SNR between the two, with more than 50% of the profiles penetrating down to 100 m above the surface, as opposed to less than 5% for MetOp and Spire NASA data. The average total monthly observations for each data product are shown in Table 1.

In summary, Figure 1(a) demonstrates that newer COSMIC-1 versions significantly outperform older versions in penetration depth. In addition, Spire NASA and MetOp data exhibit similar penetration probabilities, but NOAA-processed Spire data achieve deeper penetration compared to NASA-processed Spire data, underscoring the influence of retrieval algorithms. It is conceivable that differences between Spire NASA and Spire NOAA products in Fig. 1(a) could be attributable to the differences in the volume and sample size of available data, however, this is not found to be the case. Figure 1(b) shows the RO penetration probability for a common subset consisting of the exact sub-sample of Spire RO profiles but processed by different sources. The former is processed by the vendor and purchased by NASA as a L2 product, while the latter is processed by UCAR from the vendor-provided L1b data. Even though the same physical ROs are compared, the two products show distinctive penetration patterns below 500 meters. The penetration probabilities differ solely due to the choice of processing algorithm used for retrieving the bending angle and refractivity profiles. On the contrary, when comparing NOAA Spire profiles with the climatology of COSMIC 2021 profiles (Fig. 1(a)), both processed by UCAR, there is very little difference in the penetration probabilities. Thus, processing software appears to have a greater bearing on RO penetration loss compared to instrument hardware.

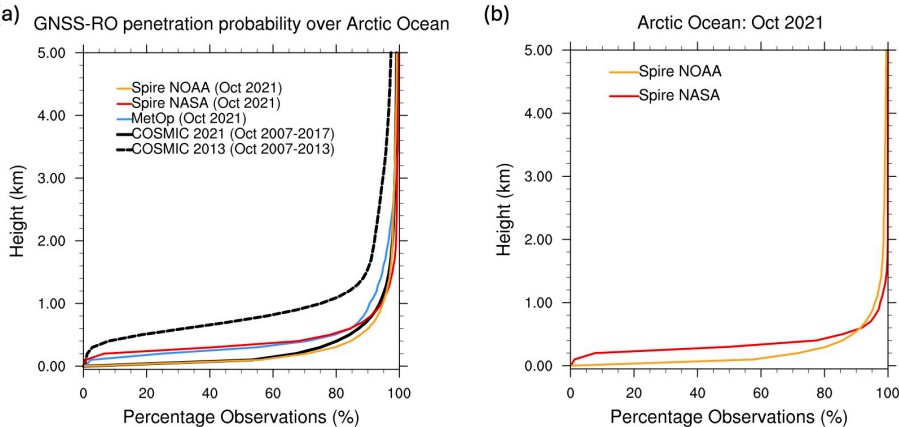

**Fig. 1** RO penetration loss as a function of altitude over the Arctic Ocean (north of 60°N) for the month of October comparing **(a)** different product versions from three major missions viz., COSMIC-1, MetOp, and Spire and **(b)** a common sub-sample from Spire NASA and Spire NOAA over the Arctic Ocean.

Similarly, Figure 2 compares the RO penetration profiles for GeoOptics, MetOp and COSMIC-1 missions. Once again, the most significant differences in RO penetration probabilities are between the old and new re-processed versions of COSMIC-1 data. Comparatively, a smaller percentage of GeoOptics profiles reach 5 km altitude, likely due to the imposed quality checks described in Sect. 2.1.1, however, a good percentage of observations (more than 50%) reach 100 m altitude, which is comparable to the 2021 re-processed COSMIC-1 product.

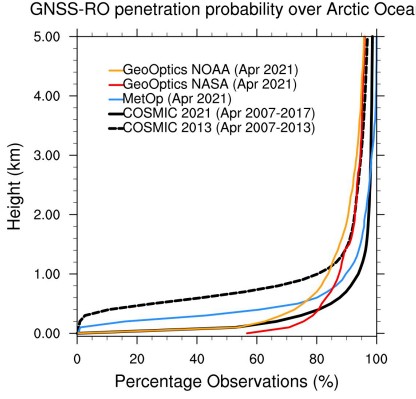

**Fig. 2** RO penetration loss as a function of altitude over the Arctic Ocean (north of 60°N) for the month of April comparing different product versions from three major missions viz., COSMIC-1, MetOp, and GeoOptics.

## 3.2 Comparison of RO penetration over the Arctic Ocean

Spatial patterns in RO penetration reveal that Spire and MetOp data have comparable penetration depths, with higher loss near coastal regions, whereas GeoOptics data exhibit the lowest penetration altitudes in regions of persistent sea ice. The top row of Figure 3 compares the minimum altitude of RO penetration over the Arctic Ocean for NASA Spire, NASA GeoOptics, and MetOp data. Spire and MetOp have similar RO penetration depth throughout the Arctic Ocean, with values dropping towards continental coastlines which is expected due to influence of topography. GeoOptics has the lowest and

highest values of minimum RO penetration altitude compared to the other two datasets, with the lows occurring over the frozen ocean in the Beaufort Sea region and to the north of Greenland, and the highs occurring over the Atlantic storm track region. A similar pattern of enhanced RO penetration loss in the storm track region was also observed in COSMIC-1 (2013 version; Ganeshan and Wu, 2015). It has been previously speculated (Ao et al., 2012; Ganeshan and Wu, 2015; Chang et al., 2022) that there is an inverse relationship between water vapor amount and RO penetration depth, with increased lower atmospheric

penetration typically observed in regions away from the tropics, specifically over the dry north pole.

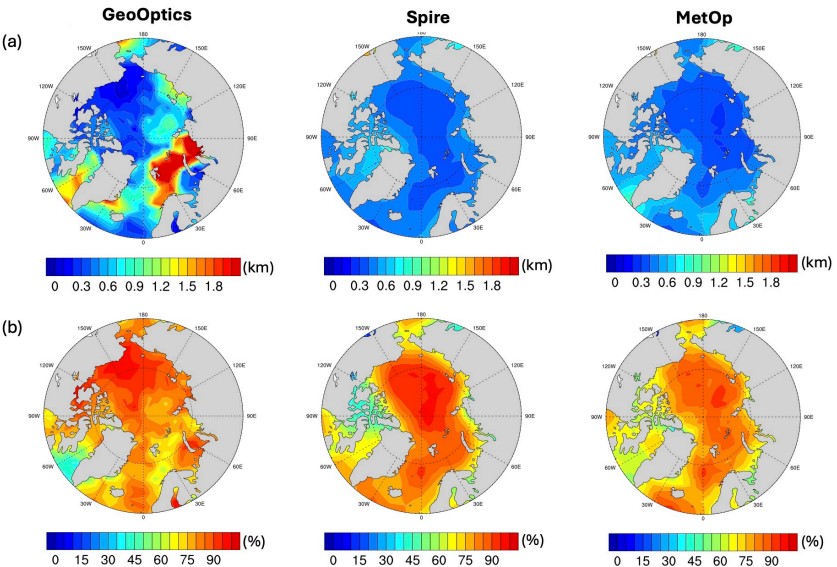

**Fig. 3** RO penetration statistics over the Arctic Ocean for December 2020 comparing GeoOptics, Spire and MetOp datasets showing **(a)** the minimum altitude of RO penetration and **(b)** the RO penetration probability at 500 m altitude.

    Previous studies (Ao et al. 2012; Ganeshan and Wu, 2015), have typically chosen a 500 m cutoff altitude to select RO profiles for retrieving the PBLH. Figure 3(b) compares the RO penetration probability at 500 m altitude between the three datasets. In general, all three products have a high fraction of RO observations (~80%) reaching 500 meters altitude. Figure 4 further compares the annual time-series of percentage of available RO observations at 500 m altitude over the Arctic Ocean.

We note a reduction of RO penetration probability for MetOp and GeoOptics during summer months which are indicative of

sensitivity to atmospheric moisture (Ao et al., 2012; Ganeshan and Wu, 2015; Chang et al., 2022). NASA Spire profiles, however, do not show a similar response to moisture, whereas NOAA Spire profiles have the same seasonality in RO penetration probability (not shown). Nevertheless, the focus of this study is winter season (November-April) during which all three datasets have similar penetration characteristics. The potential for using commercial RO data for Arctic winter PBL studies is further evaluated in the following subsection.

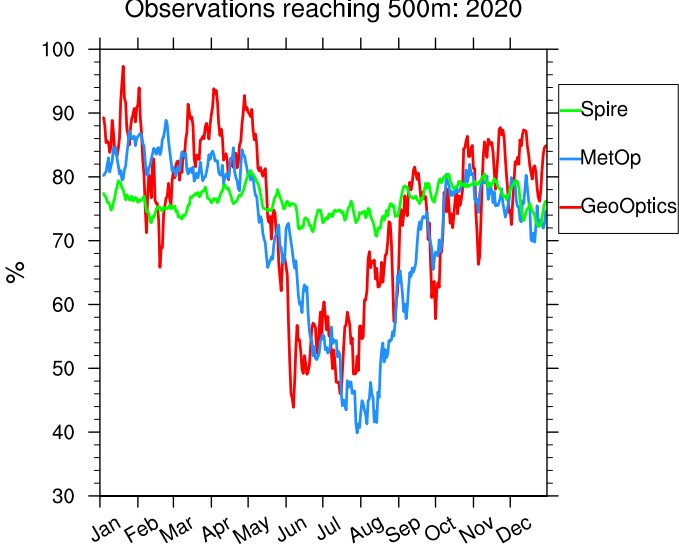

**Fig. 4** Annual time-series of percentage of RO observations reaching 500 m altitude or lower over the Arctic Ocean for the year 2020. The daily observations are smoothed using a 5-day running average filter.

### 3.3 Performance of commercial GNSS RO datasets for Arctic Ocean PBLH retrieval

Monthly mean PBLH patterns from Spire, GeoOptics, and MetOp datasets align well with MERRA-2 reanalysis (Figures 5-9). Spire's reduced spatial and seasonal variability and higher values of PBLH minima may result from its vertical smoothing processes, while GeoOptics and MetOp provide better overall representation of the Arctic PBLH.

As a first step, the cutoff altitude threshold of 500 m is chosen to select RO profiles, which has been used in previous studies (see Sect. 2.2). Ganeshan and Wu (2015) showed that the minimum refractivity gradient method works well to detect the height of PBL temperature inversions over the Arctic Ocean during winter months (November – April). Due to the lack of moisture in the atmosphere, the refractivity gradient minimum is found to be sensitive to the positive temperature gradient maxima (i.e., temperature inversions). Figures 5-7 compare the monthly RO-derived PBLH characteristics for each product during the cold season months of the year 2020 (January – April, and November – December). The adopted methodology (Ganeshan and Wu, 2015) described in Sect. 2.2, appears to work well for all three RO products, which clearly show the expected distribution of shallow PBLH over sea ice versus deeper PBLH over the Atlantic sector (MERRA-2 monthly sea ice

distributions shown in Figure 8). The extreme high values of PBLH estimates in the Atlantic Sector, seen primarily in GeoOptics and MetOp data, seem to be related to expected storm activity in this region. A seasonal evolution in the retrieved PBLH is evident in both GeoOptics and MetOp datasets with the lowest values generally observed during January, February and March (Figs. 5(a)-(c) and Figs. 7(a)-(c)), and highest values in November (Figs. 5(e) and 7(e)), which agrees with MERRA-2-derived PBLH (Fig. 9). On the other hand, NASA Spire-derived PBLH appears to have lesser spatial and seasonal variation as well as higher PBLH values over the frozen Arctic Ocean compared to the other two datasets and compared to MERRA-2, which could be because of the increased vertical smoothing applied to their bending angle product (Bowler, 2020) that may limit the effective vertical resolution of refractivity and the range of refractivity-derived PBLH values.

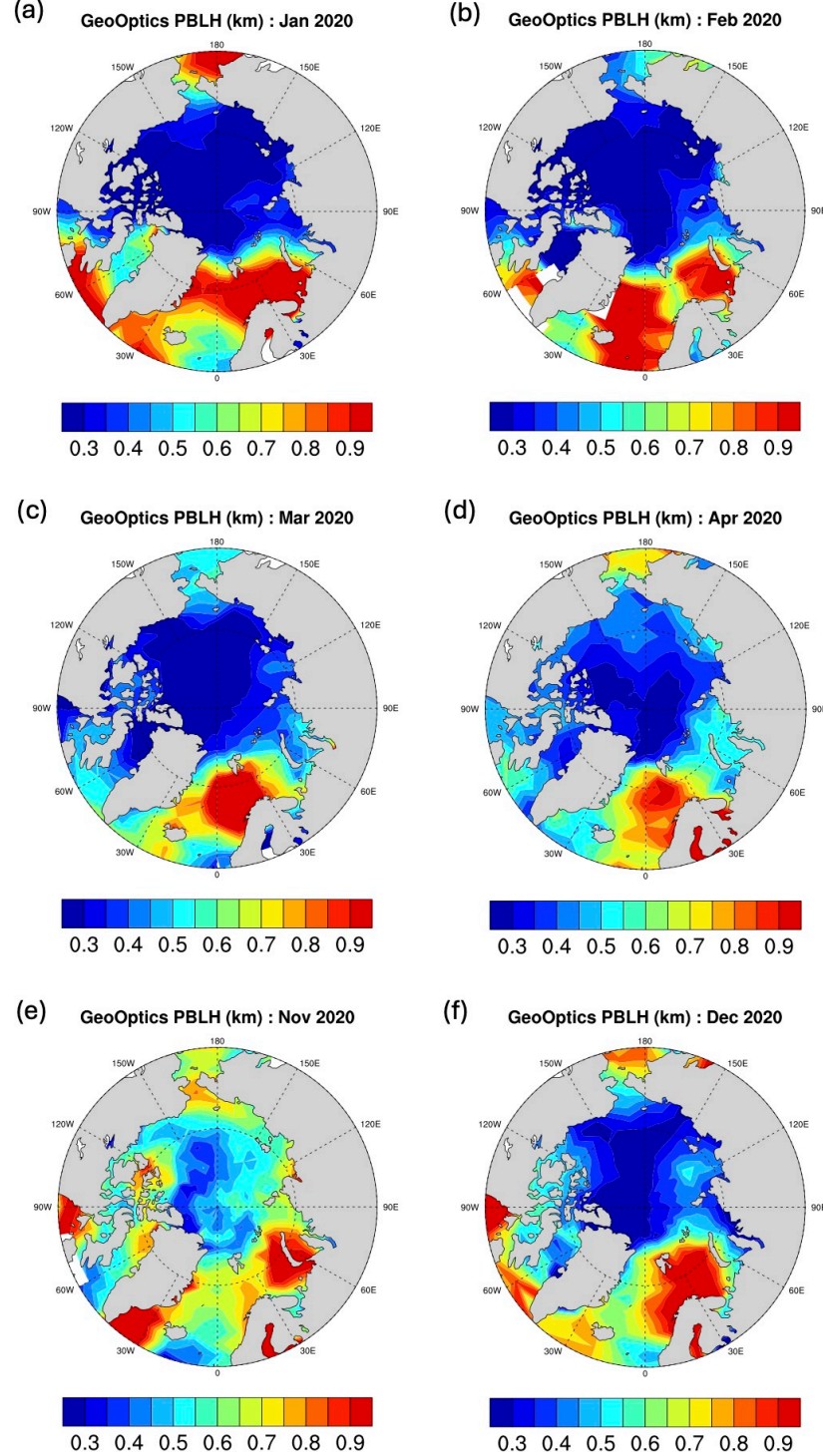

**Fig. 5** NASA GeoOptics monthly Arctic PBLH for cold season months of the year 2020.

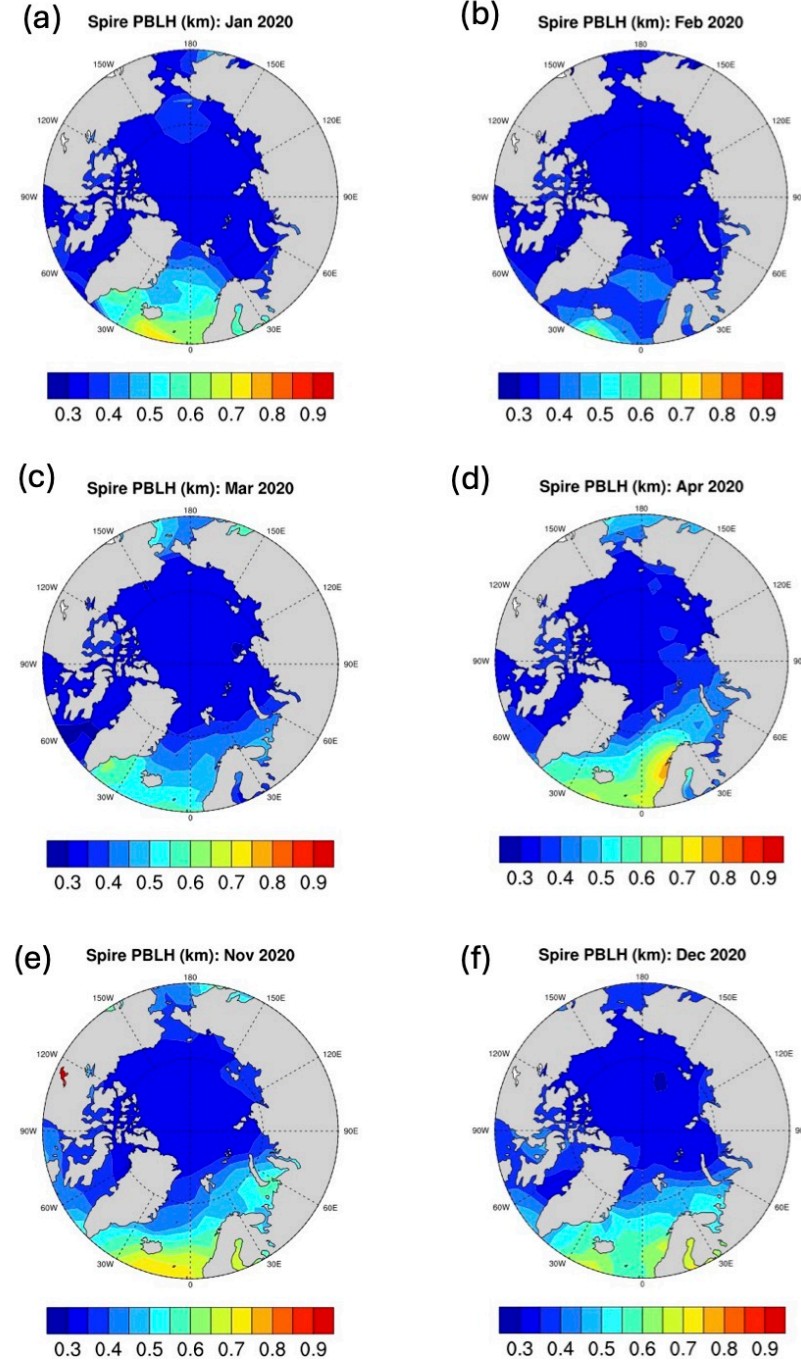

**Fig. 6** NASA Spire monthly Arctic PBLH for cold season months of the year 2020.

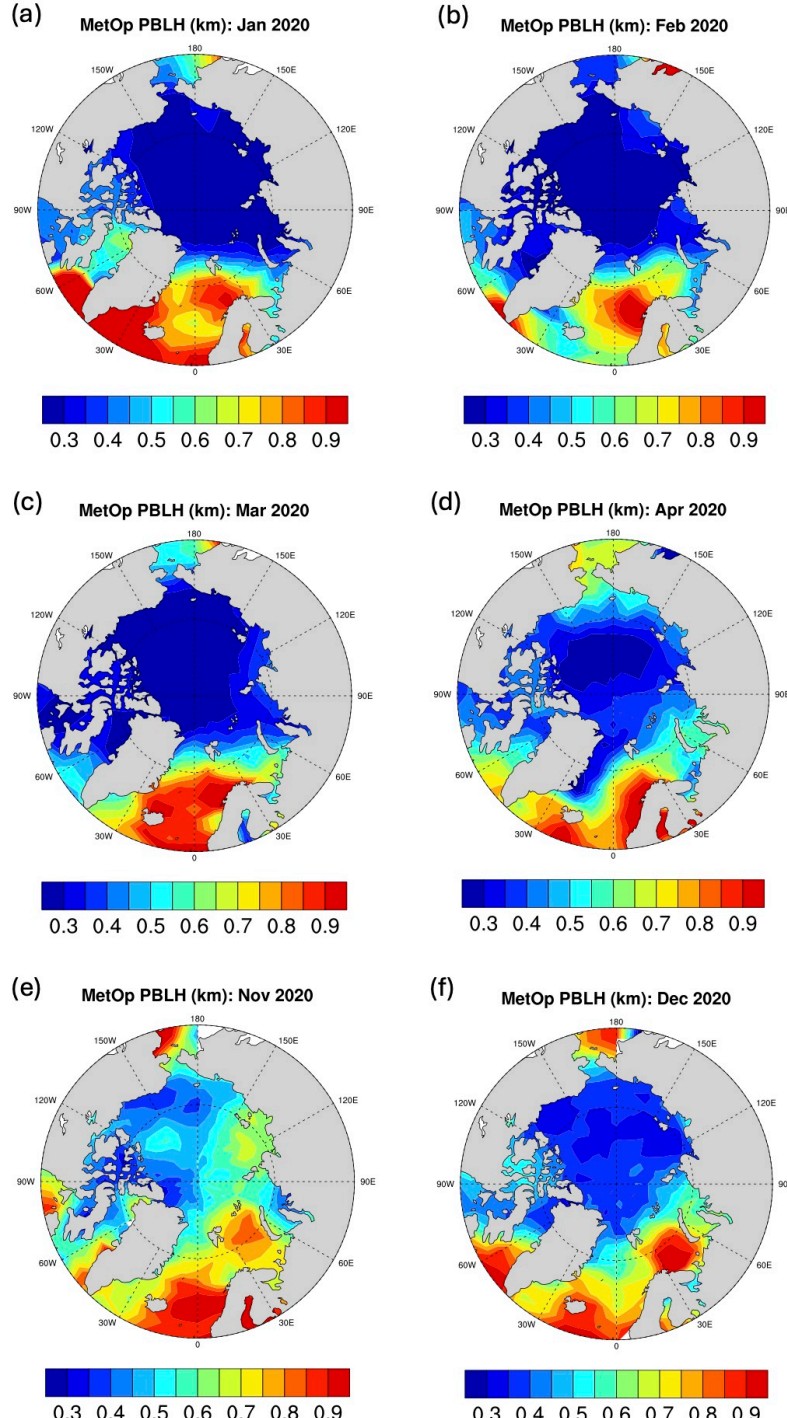

**Fig. 7** MetOP monthly Arctic PBLH for cold season months of the year 2020.

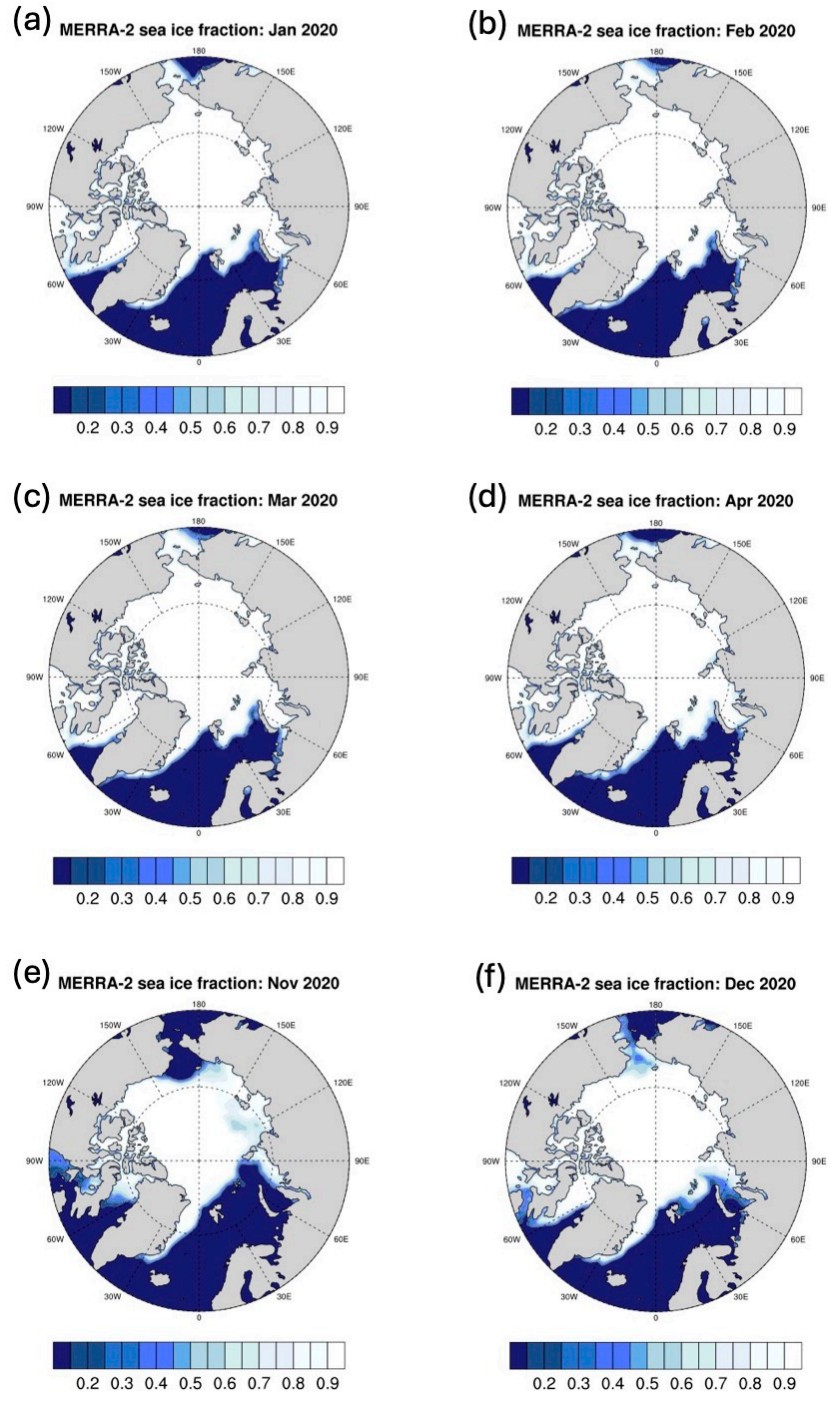

**Fig. 8** MERRA-2 monthly Arctic sea-ice fraction for cold season months of the year 2020.

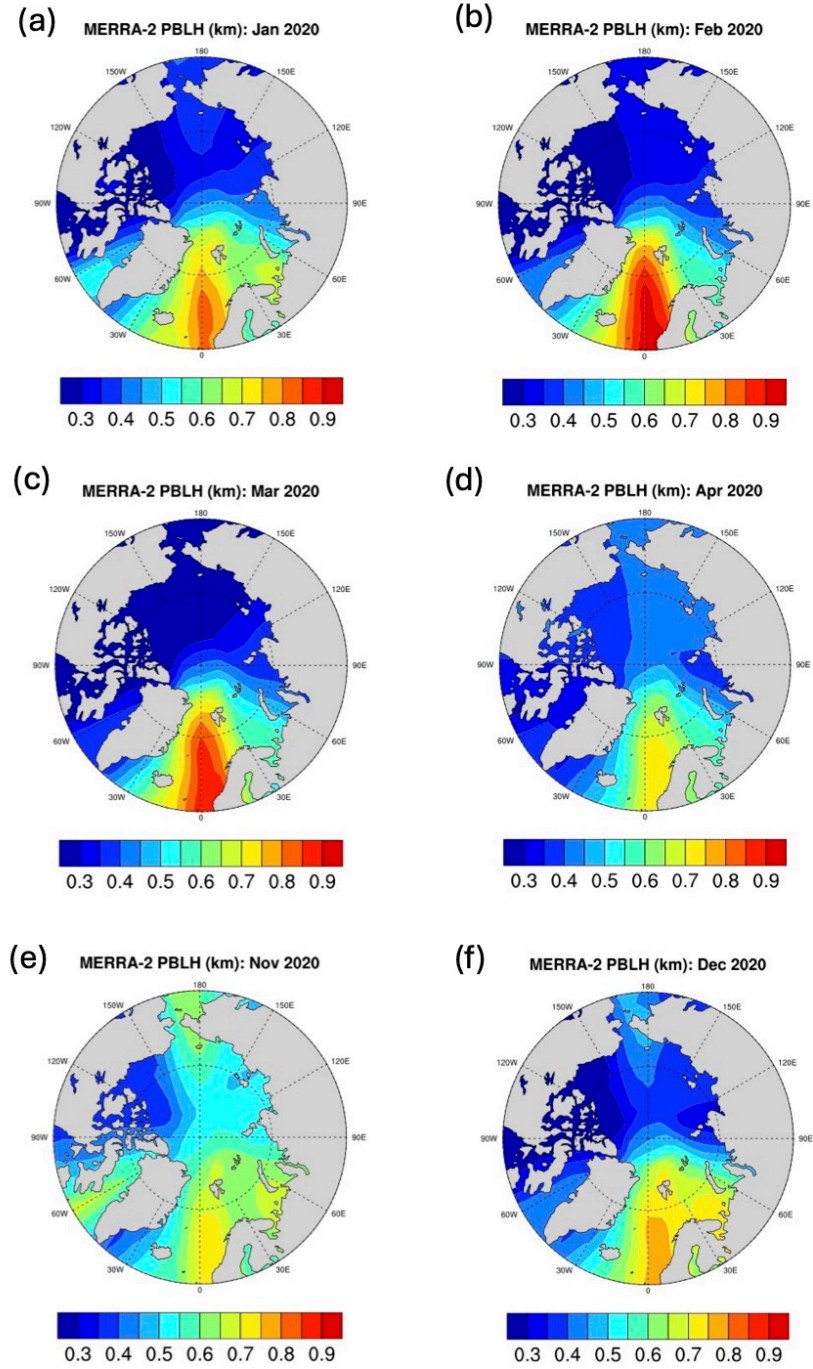

**Fig. 9** MERRA-2 monthly PBLH showing the seasonal evolution and spatial variability of Arctic PBLH for cold season months of the year 2020.

## 3.4 Sensitivity to cutoff altitude threshold

As discussed in Ganeshan and Wu (2015), a sampling bias may occur in the retrieved PBLH due to a sharp drop in available RO profiles (as seen for COSMIC-1 2013 version in Fig. 1(a)), thereby necessitating the selection of an optimal cutoff altitude threshold for minimum required RO penetration height. Although the penetration probability is much improved for commercial RO observations compared to the COSMIC-1 2013 product, with more than three factor increase in the percentage of observations reaching 500 m altitude, it is still possible that some shallow PBLs are missed. In the case of NASA-purchased Spire data, the derived PBLH values over the frozen Arctic Ocean are slightly higher compared to the other two RO datasets and MERRA-2 reanalyses (Fig. 6). It is worth investigating whether the standard 500 m cutoff altitude is suboptimal for NASA-purchased Spire data. Additionally, it is also possible that NOAA Spire refractivity profiles, which are processed using UCAR software on vendor provided L1b data, have better performance in capturing the shallow Arctic PBLH.

Figure 10 (a) shows the PBLH retrievals from NASA Spire computed using the standard cutoff altitude threshold of 500 m and a lower cutoff altitude threshold of 300 m (Fig. 10(b)). Reducing the cutoff threshold to 300 meters indeed improves retrieval of shallow PBLs over sea ice for NASA Spire data, highlighting the need for customized thresholds for different datasets (Figs. 10(a),(b)). Despite an improvement in the PBLH magnitude, the coarse spatial gradients and lacking seasonal variability (seen in Fig. 6) continue to persist for NASA Spire data even after using a lower cutoff altitude threshold (not shown). On the other hand, the NOAA-processed Spire data achieve shallower PBLs and better overall representation even with the standard 500-meter threshold (Fig. 10(c)), as evidenced by the improved spatial contrast between the frozen Arctic Ocean and open seas region (e.g., the Chukchi sea) which is missed by NASA Spire observations.  Thus, an optimal cutoff altitude threshold for representing Arctic PBLH values in NASA Spire data appears to be 300 m, however, the spatiotemporal variability in the derived PBLH is not highly impacted by cutoff altitude choice. It appears that qualitative differences in Arctic PBLH representation are mostly decided by the processing set up. In summary, both commercial RO datasets viz. Spire and GeoOptics, can satisfactorily observe the Arctic PBLH, albeit, the excessive smoothing of NASA Spire data can limit its ability to capture shallow PBLs.

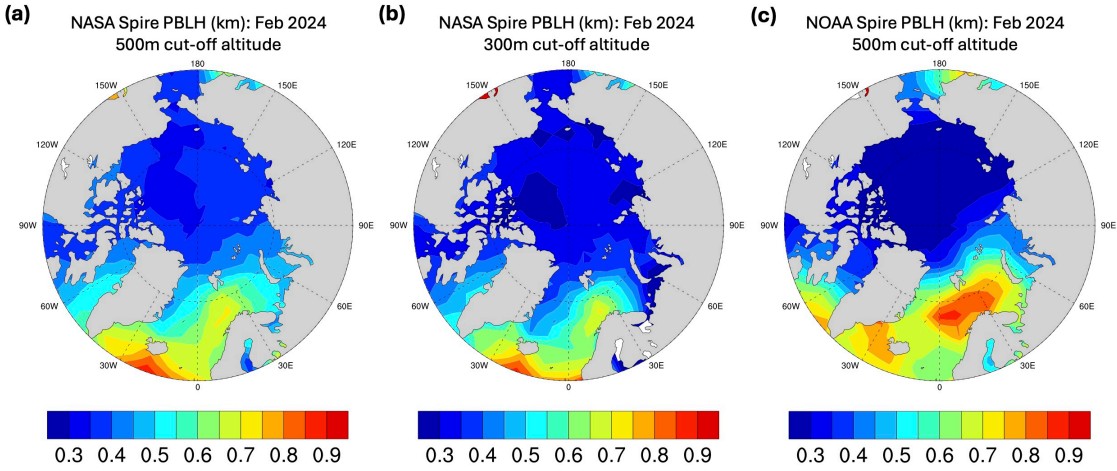

**Fig. 10** RO-derived PBLH over the Arctic Ocean for February 2024 retrieved from **(a)** NASA Spire data using 500 m cut-of altitude threshold and **(b)** NASA Spire data using 300 m cutoff altitude threshold and **(c)** NOAA Spire data using 500 m cutoff altitude threshold for minimum RO penetration depth.

## 4 Summary and Conclusions

This study demonstrates that commercial GNSS RO data from Spire and GeoOptics are valuable for Arctic PBL studies, offering penetration capabilities comparable to established missions like MetOp and COSMIC-1. Key findings include:

•    Processing algorithms have a greater impact on RO penetration depth than hardware.

•    Both commercial RO datasets provide satisfactory Arctic PBLH estimates, with NASA GeoOptics data showing better spatial and seasonal variability compared to NASA Spire data.

•    NOAA-processed Spire data outperform NASA-purchased Spire data in shallow PBL representation.

The launch of commercial GNSS RO CubeSat receivers from Spire and GeoOptics, presents an unparalleled opportunity for high-latitude PBL studies that are impacted by the loss of COSMIC-1 and the limited coverage by its successor COSMIC-2. To continue to support PBL studies in polar regions, new GNSS RO products must have sufficient lower atmospheric penetration capability, and the ability to sample shallow PBL temperature inversions that often persist in polar regions. This study attempts to provide a comparison of the penetration capability of the new commercial and other existing GNSS RO data products in the Arctic as the first step towards establishing a climate-ready, long-term continuous, dataset that can be used for Arctic PBL investigations.

It is found that the choice of processing software for retrieving neutral atmosphere bending angle and refractivity profiles has a great bearing in determining the rate of RO penetration loss in the lower troposphere, compared to factors such as instrument hardware which is consistent with previous studies (Vannah et al., 2025). Both commercial products purchased by NASA are found to have comparable lower atmospheric penetration over the Arctic Ocean to other RO climate data products

such as MetOp observations from ROM SAF and COSMIC-1 from UCAR. We identified that, on average, 80% of GeoOptics RO and Spire RO measurements could probe the Arctic troposphere as low as 500 meters. All RO datasets, with the exception of NASA-purchased Spire data, show a drop in the penetration probability during summer months signifying sensitivity to atmospheric water vapor which has been speculated in the past (Ao et al., 2012; Ganeshan and Wu, 2015; Chang et al., 2022).

The PBLH derived from the commercial RO products is agreeable with other RO datasets and reanalysis data. Despite its relatively low sampling volume as compared to Spire, the spatial pattern and seasonal evolution of Arctic Ocean PBLH are better represented by GeoOptics data. The Spire PBLH representation is seemingly improved when using NOAA-processed L2 data, suggesting sensitivity to the choice of software used for processing L1B signals. While there is some sensitivity to cutoff altitude threshold, it is predominantly the methodology used to obtain neutral atmosphere products from excess phase data that is ultimately crucial for Arctic PBLH representation. With that caveat, both Spire and GeoOptics show promising results for polar PBL studies, underscoring the importance of advancing commercial GNSS RO technology for polar climate research. Future work should focus on harmonizing processing methodologies to ensure consistent climate records.

**Acknowledgments:** This research was done in collaboration with Jet Propulsion Laboratory, California Institute of Technology under a contract with the National Aeronautics and Space Administration (80NM0018D0004) in addition to support from NASA's CSDA New Vendor Onramp Evaluation program (grant 80NSSC23K0385). The research described in this paper was partially carried out at the Jet Propulsion Laboratory, California Institute of Technology, under a contract with the National Aeronautics and Space Administration.

**Author Contributions:**

*Manisha Ganeshan* was responsible for concept, development of methodology, investigation, formal analysis, visualization, writing – original draft, writing - reviewing and editing, funding acquisition.

*Dong L. Wu* contributed to concept, development of methodology, investigation, formal analysis, writing - reviewing and editing.

*Joseph A. Santanello* contributed to concept, development of methodology, investigation, formal analysis, writing - reviewing and editing.

*Jie Gong* contributed to concept, development of methodology, investigation, formal analysis, writing - reviewing and editing.

*Chi Ao* contributed to concept, development of methodology, investigation, formal analysis, writing - reviewing and editing, funding acquisition.

*Panagiotis Vergados* contributed to concept, development of methodology, investigation, formal analysis, writing - reviewing and editing.

*Kevin J. Nelson* contributed to concept, development of methodology, investigation, formal analysis, writing - reviewing and editing.

**Competing Interests:** The contact author has declared that none of the authors has any competing interests.

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
