# Peer review of "Exploring commercial GNSS RO products for Planetary Boundary Layer studies in the Arctic"

_Atmospheric Measurement Techniques, 2024_

## Referee Comment (RC1)

1st review of the paper titled "Exploring commercial GNSS RO products for planetary boundary layer studies in the Arctic region" by Manisha Ganeshan et al.

**General comments:**

This study explored the use of commercial GNSS RO data, specifically Spire and GeoOptics, for investigating the PBL in the Arctic. It compared NASA-purchased commercial RO data with MetOp observations, highlighting the improved lower tropospheric penetration of the commercial products. However, since the NASA-purchased commercial RO data are processed by the vendors, while the MetOp data are processed by UCAR, and RO penetration probability can be related to many factors, including both hardware design and processing software, such comparison of the penetration probability between commercial and MetOp ROs may not be that scientifically interesting.

The authors also used moisture data obtained from radiosondes to investigate the relationship between moisture and MetOp RO penetration probability. The results indicate a negative correlation, significant only in March and December. During these months, the radiosonde is confined to a very limited latitude and longitude range, and the moisture over Arctic cold season is very low. This raises the question of whether such a relationship exists only in low-moisture situations. What are the authors' opinions on this matter? Does this conclusion hold for RO data processed using different algorithms? Given that the study focuses on commercial RO data, it would be beneficial to redo this investigation using the commercial data.

The last part of this study is the comparison of Arctic winter PBLH between RO and MERRA-2 reanalysis. However how is the PBLH calculate from the reanalysis data? This information is vital for the readers to understand whether the method used is appropriate and the value of the results.

Suggestions for possible revisions include:

1. As shown in Fig. 6, the typical PBLH can be as low as 300 m over the sea ice region. The specific cut-off threshold of 500 m used for selecting RO profiles may introduce biases in the resolved PBLH. Purely visualizing the resolved shallow PBLH from commercial RO as in Fig. 6 doesn't justify the choice of this threshold. A sensitivity test about the cut-off threshold is needed.
2. In addition to the map of the monthly averaged penetration probability in Fig. 5, please include the map of the monthly averaged minimum penetration depth.
3. Explain how the PBL height of MERRA-2 reanalysis was obtained/calculated, and the vertical resolution of the MERRA-2 reanalysis data.

4. Expand the PBLH study to include the Arctic summer season.

In all, I think this paper is not ready to be accepted yet and may be reconsidered after major revisions.

**Specific comments:**

1. L36-37: "improved predictability over flat surfaces compared to varying slopes". What's the meaning of this sentence? Any references support this statement?
2. L45-46: I don't understand the statement like "RO profiles over the Arctic Ocean dropped sharply …". Please rephrase it.
3. In L78, the authors mentioned that "Spire data are provided at a similar vertical grid and resolution as other GNSS RO missions", but later, when the authors tried to explain the observed lesser regional variation of PBLH from Spire compared to GeoOptics, it is mentioned that Spire data have coarser resolution due to smoothing (L250). Are these two statements contradictory to each other? What's the vertical resolution of Spire data?
4. L82: what is "the amplitude of computed phase match integral"? Any references?
5. L83: What data is considered as "at lower levels"? Below 8 km?
6. L86-89: The part is not clear to me. Has the GeoOptics data below what is called "sharp" layer been discarded by QC check? If so, does it mean the resolved PBLH later would be equal to the minimum penetration depth?
7. L90-94: Are the NOAA Spire and GeoOptics data processed by UCAR? If so, UCAR's processing starts from which level? Any useful information can we derive from the comparison between NASA and NOAA purchased commercial data?
8. L184-185: Any explanations for "missing seasonal variation in NASA Spire data, but presented in NOAA Spire data"?
9. L194: What is the vertical resolution of radiosonde observations?
10. L200-202: Please provide the results similar to Fig. 3 and 4 at 300 m, 500 m and 700 m?
11. L252-254: Clearly the cut-off height of 500 m is not sufficient to derive the shallow PBL height. I don't understand the logic of the cut-off threshold used in data analysis being allowed to be mission dependent.
12. I don't see the value of Fig. 5 e-g. May consider remove them.
13. The observed extremely high GeoOptics PBLH over the 30E to 60E sector is presented in Fig. 6. What's the explanation for this? Is it physical-related or outlier-effected?

---

## Author Comment (AC1)

**Response to Reviewer #2**

We thank the Reviewer for their useful comments and suggestions that have greatly improved our manuscript. The following outlines the changes made to our manuscript in response to the Reviewer's concerns. Reviewer comments are in italics, and responses are in regular font.

**Reviewer Comment 1:** There are multiple RO missions data set available from UCAR and ROMSAF websites, including KOMPSAT-5, PAZ, PlanetiQ, TerraSAR-X, TanDem-X, Sentinel-6a and GRACE-FO. These missions are contemporaneous with Spire and GeoOptics covering global area. They also show good penetrating capability and usefulness in PBLH application. Authors need to give explanation of the reason choosing C-1 and MetOp as the counterpart in evaluation for commercial RO.

**Response:** The Reviewer makes a good point. The primary reason for not selecting other listed satellites such as, KOMPSAT-5, PAZ, TerraSAR-X, TandDem-X, Sentinel-6a, and GRACE-FO is because RO measurements is a secondary observable. They are primarily radar imaging and radar altimeter type instruments and do not offer consistent sampling of neutral atmosphere RO as their receivers are not always turned on, as opposed to COSMIC-1 and MetOp's GRAS (Global Navigation Satellite System Receiver for Atmospheric Sounding) that are dedicated GNSS RO missions, have similar orbital heights of roughly 800 km, and offer consistent sampling of neutral atmosphere at regular fixed local time. Moreover, we did not select PlanetiQ data for this study as we are focusing on NASA-purchased commercial RO datasets with at least one year of observations.

In the revised study, we have made one change to the source of MetOp dataset. Based on the Reviewer's suggestion, we have used MetOp data from ROMSAF, which is re-processed with an updated, single software version, similar to COSMIC-1 re-processed datasets from UCAR. In the revised paper, we have made this point in section 2.1.2., Line 119 (see below): "To remove ambiguity resulting from software updates - and to ensure consistency - only those RO mission products that have been re-processed with the same software version are compared against Spire and GeoOptics."

**Reviewer Comment 2:** The radiosonde profiles from MOSAiC expedition ship campaign provide unique and valuable information on the PBLH detection in Arctic region as independent verification data. I expect to see the radiosonde derived PBLH and the direct comparison of the collocated RO-RAOB PBLH matching pairs. But in this paper the radiosonde data was only treated as source of water vapor for exploring the relationship between moisture and RO penetration. Consider the very limited geographic coverage of the ship campaign, does the conclusion cannot be achieved by using water vapor profile from climate model data, like MERRA-2 or ERA-5?

**Response:** As the Reviewer points out, the radiosonde data are limited spatially and not ideal for comparing against GNSS RO-derived measurements which have a coarse horizontal resolution (100-200 km). As a result, the radiosonde profiles are not used for RO-RAOB PBLH

comparisons. Instead, we use MERRA-2 reanalyses for comparing RO-derived PBLH. We agree with the Reviewer that the Arctic PBLH comparison should be the main focus of this paper, and we have therefore removed the section comprising the analysis of water vapor and RO penetration probability. MOSAiC radiosonde observations are no longer used as a dataset in the revised paper.

**Reviewer Comment 3:** As far as I know, the MERRA-2 reanalysis provides two PBLH values. One is calculated based on the total eddy diffusion coefficient of heat  $(K_h)$ , and the other one is estimated using the bulk Richardson number method. The PBLH calculated from  $K_h$  is usually higher than the one from bulk Richardson number in most regions. It would be interesting to check the difference over the Arctic area and validate it using RO derived PBLH. However, the paper did not provide much description on the MERRA-2 PBLH. Reader don't even know how the PBLH was extracted from MERRA-2 reanalysis.

**Response:** The version of the GEOS model used in MERRA-2 includes two PBL parameterization schemes, viz. Lock scheme which is activated for unstable PBLs and the Louis scheme which is activated for stable PBLs. The model PBL depth is defined as the model level where the eddy heat diffusivity coefficient (KH) value falls below 2 m2 s-1 threshold. At a given time, only one PBL depth value is calculated by the model, either by the Lock scheme or by the Louis scheme. Both schemes use different methods of estimating eddy diffusivity coefficients, and therefore PBL heights. We have included a description of the schemes and a discussion of their relevance for the Arctic Ocean in the revised paper. Please, see Section 2.2.

**Reviewer Comment 4:** In the PBLH deriving method section (2.1.3), the method of first minima of the refractivity gradient to exceed -40 N-unit km-1 and the 500 m threshold for RO cut-off height are chosen without justification. A sensitivity study for the threshold, comparison of different methods (minimum gradient, wavelet covariance transformation etc.) and variables (refractivity based, bending angle based) are recommended, according to the discussion for figure 6(b) and 6(c).

**Response:** The goal of this study is to compare the PBL heights derived from commercial RO datasets using the previously established and validated methodology (Ganeshan and Wu 2015) for Arctic Ocean. This methodology is found to work well for cold season (Nov-Apr) months over the Arctic. Radiosonde observations from the SHEBA campaign were used to demonstrate that low specific humidity during the cold season months led to a heightened sensitivity of refractivity to temperature gradients (Ganeshan and Wu 2015). The radiosonde data show an empirical relationship between the height of the first local minima in refractivity gradient and the Arctic PBL temperature inversion height. Furthermore, the methodology is found to yield reasonable monthly mean PBLH values when applied to COSMIC-1 observations (Ganeshan and Wu 2015). Thus, in this study, the same methodology is adopted. The sensitivity to cut-off altitude threshold, however, is discussed in section 3.4.

**Reviewer Comment 5:** The seasonal variation of RO penetration probability is displayed and discussed, whereas the more important seasonal variation of PBLH was not provided. In my opinion, a big picture of PBLH in north pole region is desirable (seasonal variation, diurnal circle if any, longitudinal variability related to the Atlantic Ocean current and sea ice

distribution etc.) in the section 1.1, then a statement of how commercial RO can improve the understanding in section 3.3.

**Response:** This is a good suggestion. We have included in the revised paper monthly mean RO-derived PBLH for six months of the year, showing seasonality and spatial variability due to the distribution of sea ice and open water (Figures 5-9). Since the RO-derived PBLH retrieval only works well for cold season months as described in Ganeshan and Wu (2015), we only estimate the seasonal cycle for November to April.

**Reviewer Comment 6:** In figure 1, the different penetration probability of Spire NOAA and Spire NASA may contributed by the sample noncoincidence, because Spire NOAA is a small subset of Spire NASA (~3500 out of ~12000 in one day). Whereas for GeoOptics, NOAA and NASA are basically covering the same observations. Therefore the explanation of the discrepancy of orange/red lines may be completely different. Since the paper introduced NASA purchased commercial RO, which is processed by vendor, and NOAA purchased commercial RO, which is processed by UCAR, it's ideal to derive PBLH using both NASA and NOAA commercial RO, to help understanding the factors affecting RO penetration.

**Response:** This is a great recommendation. In the revised paper, we have included a figure comparing the exact same subset of Spire radio occultations purchased by NASA and by NOAA. Indeed, the difference in the penetration statistics are due to differences in processing software as shown in Figure 2(a). We further drive home this point by including a figure (Fig. 2(b)) showing the similarity in penetration statistics between two different sources of radio occultations, viz. COSMIC-2 and Spire NOAA, over the tropics that are processed by the same software.

---

## Author Comment (AC4)

We thank the Reviewer for their useful comments and suggestions that have greatly improved our manuscript. The following outlines the changes made to our manuscript in response to the Reviewer's concerns. Reviewer comments are in italics, and responses are in regular font.

**General Reviewer Comment:** *The last part of this study is the comparison of Arctic winter PBLH between RO and MERRA-2 reanalysis. However how is the PBLH calculate from the reanalysis data? This information is vital for the readers to understand whether the method used is appropriate and the value of the results.*

**Response:** The version of the GEOS model used in MERRA-2, includes two PBL parameterization schemes, viz. Lock scheme which is activated for unstable PBLs and the Louis scheme which is activated for stable PBLs. The model PBL depth is defined as the model level where the eddy heat diffusivity coefficient ($K_H$) value falls below 2 $m^2s^{-1}$ threshold. At a given time, only one PBL depth value is calculated by the model, either by the Lock scheme or by the Louis scheme. Both schemes use different methods of estimating eddy diffusivity coefficients, and therefore PBL heights. We added a description of the schemes and a discussion of their relevance for the Arctic Ocean is included in the revised paper. Please, see Section 2.2.

**Reviewer Comment 1:** *As shown in Fig. 6, the typical PBLH can be as low as 300 m over the sea ice region. The specific cut-off threshold of 500 m used for selecting RO profiles may introduce biases in the resolved PBLH. Purely visualizing the resolved shallow PBLH from commercial RO as in Fig. 6 doesn't justify the choice of this threshold. A sensitivity test about the cut-off threshold is needed.*

**Response:** In the revised paper, the sensitivity to cut-off altitude threshold is extensively discussed in sections 2.1.3 and 3.4. For most RO datasets, the standard 500m cut-off altitude threshold works well. It is found that Spire NASA data perform better when a lower cut-off altitude of 300 m is used instead of the standard 500 m threshold. Please see section 2.1.3, section 3.4, and Fig. 11 of the revised manuscript.

**Reviewer Comment 2:** *In addition to the map of the monthly averaged penetration probability in Fig. 5, please include the map of the monthly averaged minimum penetration depth.*

**Response:** We agree with the reviewer. This is a good suggestion. We have included a map of the monthly average minimum penetration depth, and revised Figure 3 to accommodate the inclusion. Please, see revised Fig. 3 in the main manuscript.

**Reviewer Comment 3:** *Explain how the PBL height of MERRA-2 reanalysis was obtained/calculated, and the vertical resolution of the MERRA-2 reanalysis data.*

**Response:** Thank you for the suggestion. We have included text to describe the parameterization schemes used for computing MERRA-2 PBL height and their relevance for the Arctic Ocean. Please, see Section 2.2.

The vertical grid of MERRA-2 is based on terrain-following sigma coordinate system, wherein the exact model level height is a function of the surface pressure. In general, the first model level is

around 50 m above surface and the spacing is approximately 100 m within the lowest five model levels. **This information is now included in section 2.2 of the main manuscript as well**.

**Reviewer Comment 4:** *Expand the PBLH study to include the Arctic summer season.*

**Response:** The methodology used to compute the PBLH in this study is based on Ganeshan and Wu (2015) which is a validated RO technique for PBLH estimation over the Arctic Ocean during winter months (Nov-Apr). This technique works well when the specific humidity is low, and the refractivity profile is mainly sensitive to temperature gradients. In the revised paper, we have expanded on the PBLH study to all cold season months (i.e. Nov-Apr).

**Specific comments:**

**1. L36-37:** "*improved predictability over flat surfaces compared to varying slopes*".

What's the meaning of this sentence? Any references support this statement?

**Response:** This refers to the fact that GNSS RO has a low horizontal resolution (100-200 km) and therefore is expected to perform well over topographically homogenous surfaces, which includes flat surfaces (sea ice, open ocean) compared to sharp varying slopes (e.g. coastal land mass areas). The statement has been rephrased as follows: "improved performance over flat surfaces (sea ice, open ocean) compared to sharp varying slopes (land mass),…". Please see section 1.1, Line 42, of revised manuscript.

**2. L45-46:** *I don't understand the statement like "RO profiles over the Arctic Ocean dropped sharply …". Please rephrase it.*

**Response**: This statement has been rephrased in the revised manuscript (section 1.1, Line 53) as follows:

"From the analysis of 8 years of COSMIC-1 data, it was found that availability of RO profiles over the Arctic Ocean reduced significantly at tangent heights below 1km, which introduces a sensitivity of the retrieved PBL height to the choice of the cut-off altitude, or minimum RO penetration depth, used for profile selection".

**3. In L78**, *the authors mentioned that "Spire data are provided at a similar vertical grid and resolution as other GNSS RO missions", but later, when the authors tried to explain the observed lesser regional variation of PBLH from Spire compared to GeoOptics, it is mentioned that Spire data have coarser resolution due to smoothing (L250). Are these two statements contradictory to each other? What's the vertical resolution of Spire data?*

**Response:** Spire data are indeed provided at a similar vertical grid as other GNSS RO missions, however, NASA Spire bending angle profiles are excessively smoothed prior to the refractivity retrieval, thus reducing their effective vertical resolution (i.e. loss of information due to smoothing).

**4. L82:** *what is "the amplitude of computed phase match integral"? Any references?*

**Response:** The methodology for retrieving GeoOptics neutral atmosphere profiles and their quality control is described in the "GeoOptics Processor for Radio Occultation (GeoPRO) User Guide" which was provided to NASA by the vendor. Phase matching (Jensen et al., 2004) in RO processing is a wave optics technique designed to extract the full information from the received wave field. It is conceptually and practically simpler than other wave optics techniques, while producing a number of useful diagnostics and additional features. It has often been used in detailed analysis of individual occultations and produces radio holographic images of each occultation which are extremely useful in diagnosing signal or processing issues and can even reveal new information. As part of the quality control, profiles are cut-off at low altitude when the phase match amplitude falls below a certain threshold.

**5. L83:** *What data is considered as "at lower levels"? Below 8 km?*

**Response:** Yes, the levels below 8km are flagged as bad quality if the blanket criteria check is failed. This is clarified in the revised manuscript (section 2.1.1, Line 86).

**6. L86-89:** *The part is not clear to me. Has the GeoOptics data below what is called "sharp" layer been discarded by QC check? If so, does it mean the resolved PBLH later would be equal to the minimum penetration depth?*

**Response:** No, if the QC check is passed at any altitude below "sharp layers", the data are not discarded. Each profile is evaluated individually to determine the minimum penetration depth ascertained by the lowest above-surface level with a "good" quality flag. If a "sharp" PBL inversion layer with poor QC flag exists above the minimum penetration depth, then this is not disregarded. We rephrased this part and better explained it in the revised manuscript. See, Section 2.1.

**7. L90-94:** *Are the NOAA Spire and GeoOptics data processed by UCAR? If so, UCAR's processing starts from which level? Any useful information can we derive from the comparison between NASA and NOAA purchased commercial data?*

**Response:** Yes, the NOAA Spire data are processed by UCAR from purchased L1b data. The NASA-purchased Spire data are processed to level 2 by the vendor.

**8. L184-185:** *Any explanations for "missing seasonal variation in NASA Spire data, but presented in NOAA Spire data"?*

**Response:** Differences in seasonal variability between NASA Spire and NOAA Spire data are evident because the two datasets are processed by different methodologies.

**9. L194:** *What is the vertical resolution of radiosonde observations?*

**Response:** Radiosonde observations are no longer used in this study.

**10. L200-202:** *Please provide the results similar to Fig. 3 and 4 at 300 m, 500 m and 700 m?*

**Response:** Figures 3 and 4 are no longer part of the analysis. The water vapor and RO penetration probability relationship will be explored in a follow-on study.

**11. L252-254:** *Clearly the cut-off height of 500 m is not sufficient to derive the shallow PBL height. I don't understand the logic of the cut-off threshold used in data analysis being allowed to be mission dependent.*

**Response:** The cut-off altitude or minimum required RO penetration depth, in some sense, is a first guess estimate of the expected typical height of the PBL. A sampling bias may occur in the retrieved PBLH due to a sharp drop in available RO profiles, thereby necessitating the selection of an optimal cut-off altitude threshold for minimum RO penetration depth. While the standard cut-off altitude of 500m has been regarded as sufficient for deriving refractivity-based PBLH from COSMIC-1 RO observations in the Arctic, it is carefully examined in this study for use with different RO datasets. It appears that the 500 m cut-off altitude when applied to NASA GeoOptics and ROMSAF MetOp data is sufficient for obtaining a realistic representation of the shallow Arctic PBLH. However, in the case of NASA Spire data, the derived PBLH values are slightly higher compared to the other two RO datasets and MERRA-2 reanalyses (Fig. 6 in revised manuscript). This is because the percentage of available NASA Spire RO profiles drops significantly going from 400 to 300 m and then from 300 to 200 m (Figure 10 in revised manuscript), which could potentially lead to a positive bias in the retrieved PBLH values when the standard cut-off altitude of 500 m is chosen. No such sharp drop is seen for GeoOptics and MetOp datasets. Moreover, a similar comparison with the NOAA Spire product shows that this sharp rate of decline only exists in the NASA Spire data. As a result, the PBLH retrievals from NASA Spire data are recomputed using a lower cut-off altitude threshold of 300 m, and the resulting PBLH values are found to be significantly lower and in better agreement with other datasets (see Fig. 11 in revised manuscript). However, the spatial patterns and seasonality are not impacted by the choice of cut-off altitude threshold. In summary, an optimal cut-off altitude threshold for RO products can be chosen based on their rate of RO penetration decline within the PBL. However, the impact of cut-off altitude threshold is limited to simply an improvement in the magnitude of the retrieved PBLH, and no strong sensitivity is observed to spatiotemporal patterns. This is explained in section 3.4 of the revised paper.

**12.** *I don't see the value of Fig. 5 e-g. May consider remove them.*

**Response:** These figures and discussion are no longer part of the new manuscript.

**13.** *The observed extremely high GeoOptics PBLH over the 30E to 60E sector is presented in Fig. 6. What's the explanation for this? Is it physical-related or outlier-effected?*

**Response:** The extreme high values of GeoOptics based PBLH in the Atlantic Sector seems to be related to the high minimum penetration altitude in this region (seen in Fig. 3). This is noted in the revised manuscript in section 3.3, Line 234.

**Author References:**

Jensen, A. S., Lohmann, M. S., Nielsen, A. S., and Benzon, H.-H.: Geometrical optics phase matching of radio occultation signals, Radio Science, 39, n/a-n/a, https://doi.org/10.1029/2003rs002899, 2004.

---

## Referee Report (RR1)

**2nd Review of "Exploring commercial GNSS RO products for PBL studies in the Arctic Region" by Ganeshan et al.**

I appreciate the authors' efforts in addressing the reviewers' comments and suggestions, including adding a section on the sensitivity tests of cut-off thresholds for deriving PBLH. However, several key concerns remain inadequately addressed in the responses and/or revised manuscript:

1. Since this study incorporates multiple datasets, and given that the authors acknowledge the impact of processing algorithms on the RO penetration depth, it is essential to provide clear details in Section 2. Please specify which data centers processed the data, the data availability period, the average daily RO counts over the Arctic Ocean, data version, processing modes (e.g., real-time, postprocessed, or reprocessed), and other relevant information. A summary table could be helpful here to enhance clarity.

2. Consider modifying the title from "Arctic region" to "Arctic Ocean" to more accurately reflect the study's geographic focus.

3. In Fig. 1, the authors compared the commercial dataset purchased by NASA and NOAA for the same month but not for a common dataset, which could result in sampling differences. It may be premature to attribute the observed differences in the penetration probability solely to the difference of the processing algorithms (Line 170-171). A similar concern applies to Fig. 2b, where the Spire and COSMIC-2 datasets over 30S-30N are compared. The sampling difference between these two missions may be significant due to their distinct RO count distributions with latitudes. The authors may consider using a collocated Spire-COSMIC-2 dataset to replot this figure. It could minimize the impact of sampling difference and provide more robust results. Additionally, if these statistics include the regions beyond the tropical ocean, terrain effects should be accounted for when generating this figure.

4. Fig. 4 shows that the daily RO counts reaching below 500 m for GeoOptics over the whole Arctic Ocean range from a few to 25. Such amount and variability raise concerns about whether GeoOptics data are sufficient to reliably capture the spatial variability of PBLH month by month. Could the authors comment on the reliability of GeoOptics data for deriving monthly PBL structure and variability?

5. The NASA Spire-derived PBLH exhibits lesser spatial and seasonal variability compared to the other two datasets, which the authors attribute to highly smoothed vertical RO retrievals. However, NOAA Spire RO data are not similarly smoothed. Why not present the PBLH derived NOAA Spire data to substantiate this explanation?

6. The last paragraph of the summary lacks scientific accuracy. The discussion is rather weak without reliable justification. For instance, could the author define what constitutes a "smooth" versus a "dramatic" change in the decline rate of RO penetration?

---

## Author Response (AR2)

**Response to Reviewer 1**

We thank the Reviewer for their useful comments and suggestions that have greatly improved our manuscript. The following outlines the changes made to our manuscript in response to the Reviewer's concerns. Reviewer comments are in italics, and responses are in regular font.

*I appreciate the authors' efforts in addressing the reviewers' comments and suggestions, including adding a section on the sensitivity tests of cut-off thresholds for deriving PBLH. However, several key concerns remain inadequately addressed in the responses and/or revised manuscript:*

**Response:** We thank the Reviewer for their first and second rounds of review, and for their useful insights. We have now included a table describing all RO datasets used in our study in the revised manuscript upon the Reviewer's recommendation.

*1. Since this study incorporates multiple datasets, and given that the authors acknowledge the impact of processing algorithms on the RO penetration depth, it is essential to provide clear details in Section 2. Please specify which data centers processed the data, the data availability period, the average daily RO counts over the Arctic Ocean, data version, processing modes (e.g., real-time, postprocessed, or reprocessed), and other relevant information. A summary table could be helpful here to enhance clarity.*

**Response:** This is a very helpful suggestion. In the revised manuscript, we now include a table with the requested information (Table 1).

*2. Consider modifying the title from "Arctic region" to "Arctic Ocean" to more accurately reflect the study's geographic focus.*

**Response:** Thank you for the suggestion. We have now modified the title to read "Exploring commercial GNSS RO products for Planetary Boundary Layer studies in the Arctic".

*3. In Fig. 1, the authors compared the commercial dataset purchased by NASA and NOAA for the same month but not for a common dataset, which could result in sampling differences. It may be premature to attribute the observed differences in the penetration probability solely to the difference of the processing algorithms (Line 170-171). A similar concern applies to Fig. 2b, where the Spire and COSMIC-2 datasets over 30S-30N are compared. The sampling difference between these two missions may be significant due to their distinct RO count distributions with latitudes. The authors may consider using a collocated Spire-COSMIC-2 dataset to replot this figure. It could minimize the impact of sampling difference and provide more robust results. Additionally, if these statistics include the regions beyond the tropical ocean, terrain effects should be accounted for when generating this figure.*

**Response:** We agree that differences between the penetration probabilities of NASA and NOAA GeoOptics data need to be evaluated for a common overlapping subset of RO profiles, as we did for Spire data in Fig. 1(b) of the revised manuscript. However, we think it would be repetitive to

have a similar figure, hence we have removed the speculative statement concerning GeoOptics from our revised manuscript.

Figure 2(b) in the original manuscript comparing COSMIC-2 and Spire data over the tropics was only considering profiles over the tropical ocean (not land regions). Even though they were not collocated samples of RO profiles, the similarity in their penetration probability curve suggests that they are likely to have even better agreement when comparing a common collocated subset of ROs. However, this figure no longer appears in our revised manuscript. Instead, the same point is made by comparing Spire NOAA with COSMIC-1 data in Figure 1(a).

*4. Fig. 4 shows that the daily RO counts reaching below 500 m for GeoOptics over the whole Arctic Ocean range from a few to 25. Such amount and variability raise concerns about whether GeoOptics data are sufficient to reliably capture the spatial variability of PBLH month by month. Could the authors comment on the reliability of GeoOptics data for deriving monthly PBL structure and variability?*

**Response:** This is a good point. The revised manuscript now includes a table with the average monthly RO count for each satellite product over the Arctic Ocean. NASA GeoOptics has the least number of profiles, averaging roughly 754 per month. Given that nearly 80% of these profiles (~600) reach the altitude of 500 m, we think there are enough observations for deriving monthly mean PBLH maps.

*5. The NASA Spire-derived PBLH exhibits lesser spatial and seasonal variability compared to the other two datasets, which the authors attribute to highly smoothed vertical RO retrievals. However, NOAA Spire RO data are not similarly smoothed. Why not present the PBLH derived NOAA Spire data to substantiate this explanation?*

**Response:** The NOAA Spire data are a near-real-time product, and not available for our entire study period. However, we have used an example to show the superior performance of NOAA Spire derived PBLH in comparison to NASA Spire derived PBLH in Figure 10 of the revised manuscript.

*6. The last paragraph of the summary lacks scientific accuracy. The discussion is rather weak without reliable justification. For instance, could the author define what constitutes a "smooth" versus a "dramatic" change in the decline rate of RO penetration?*

**Response:** The revised manuscript does not show the rate of decline of RO penetration (Fig. 10 in old manuscript) as it was no longer relevant to our conclusions. We have removed the discussion related to the drastic decline in RO penetration observed for NASA Spire data.

**Response to Reviewer 2**

We thank the Reviewer for their useful comments and suggestions that have greatly improved our manuscript. The following outlines the changes made to our manuscript in response to the Reviewer's concerns. Reviewer comments are in italics, and responses are in regular font.

*With the PBL finally receiving its deserved attention, this presents an interesting assessment of different instruments and data streams for PBLH detection in Arctic regions. I do however have several major comments. This is primarily related to my in-depth knowledge of Metop data (though that information is also publicly availablefrom the ROM SAF website), but also due to the presentation of results, use of figures, omission of some ROM SAF data.*

**Response:** We have carefully considered all comments by the Reviewer. Particularly, we have taken into account the different nuances of available MetOp data streams, and have chosen the most relevant product for our study. We thank the Reviewer for bringing this to our attention.

**Major Comments:**

*This reprocessed ICDR ROM SAF data represents a rather old processing setup, developed sometime in 2017 and frozen in time (it is an ICDR data set, as also mentioned on the ROM SAF website). The latest, Metop NRT data, is using a much improved processing setup. E.g. ICDR's lowest altitude reached for refractivity processing, using the first week of 2024, is on average 1.5km, while NRT's is 0.87km. The distributions also look very different. And, if you were to split this up further,you'll also find a different setting vs. rising distribution (due to the use of raw sampling tracking on Metop, not a "full open loop"). Thus, what you primarily see in Metopdata is similar to what you see in the 2 COSMIC data streams (Figure 1). Improved processing is available with more recent data sets. Maybe, to show that Metop datahas similar penetration improvements, it might be worth to include the ROM SAF NRT data stream too? Or, at least make it much clearer that the Metop data is notrepresentative of the current processing.*

**Response:** We thank the Reviewer for sharing insightful information on the available MetOp data products.
Figure R1 (left panel) compares MetOp NRT versus MetOP ICDR for reference, and there is indeed improvement in RO penetration with the former.
Our study is focused on evaluating commercial RO datasets against established climate data records using stable, long-term observations. This goal prevents us from using NRT products as they lack consistency due to frequent software updates. This is explained in Line 114 in the revised manuscript (see below):
"Although MetOP near-realtime (NRT) product from ROM SAF has more advanced processing setup with improved lower tropospheric penetration, the goal is to compare with a consistent climate record to avoid ambiguities resulting from frequent software updates. Therefore, the ICDR data are used in this study."

Moreover, based on the Reviewer's recommendation, we additionally inspected the differences between rising and setting occultations for the MetOp ICDR dataset (right panel of Figure R1), and found that setting occultations have improved RO penetration, which is consistent with the findings of Innerkofler et al. (2023). As a result, we have only used setting occultations in the revised study, and the resulting penetrations statistics are much improved and comparable to other RO datasets.

[Figure]

**Fig. R1** RO penetration loss as a function of altitude over the Arctic Ocean (north of 60°N) for the month of October 2021 comparing **(left)** MetOp NRT and MetOP ICDR products and **(right)** rising and setting occultations from MetOp ICDR product.

*Is the nine-point local smoothing applied in longitude, or latitude, or both? And if used in longitude, given that the difference between 2 longitude points is getting very small the closer one gets to the pole, what is the impact of this? And wouldn't it in general be better to apply this smoothing to all data sets, so that one compares likes with likes? And use a smoothing over a fixed area, not fixed latitude/longitude?*

**Response:** We thank the Reviewer for pointing out the disadvantages of using grid-point based smoothing for polar regions. In the revised manuscript, we have produced new figures without using any smoothing. The results are not changed much.

*P5L160: also due to the viewing geometry, as rising is more difficult to track, and sometimes even has a different antenna.*

**Response:** This line (Line 178 in revised manuscript) has been modified to include the differences due to viewing geometry as suggested by the Reviewer.

"Penetration loss can also be different for measurements from the same instrument due to the viewing geometry, as rising occultations can be more difficult to track (Innerkofler et al. 2023), as well as due to inherent disparity in excess phase computations and bending angle retrieval algorithms".

*P5L163: as stated above, this applies in the same manner to Metop data.*

**Response:** Agreed. This has been noted in Line 114 of the Revised manuscript.

*P8L216: It might be instructive to mention that reductions in Metop/GeoOptics are visible, as they show all data from those satellites. Spire on the other hand likely has a contract to fulfil, and can select from more satellites.*

**Response:** This statement corresponds to Figure 4 in the old manuscript which is no longer shown in the revised manuscript. Instead, Table 1 shows the average monthly available observations for each RO product, and Spire has the maximum count. Figure 4 in the revised manuscript shows the annual time-series of the **percentage** of available RO observations at 500 meter altitude. A seasonality is clearly seen for GeoOptics and MetOp which is related to the sensitivity of RO penetration to atmospheric moisture. This is observed for all RO products (COSMIC-1, NOAA Spire, NOAA GeoOptics), however it is not seen in NASA Spire data, which is duly reported in Line 244 of the revised manuscript.

*P9L229: There are 5 figures covered in this one paragraph. Each has 6 sub-figures. I assume this can be shortened to 2 figures, and only pointing out differences withrespect to a few representative ones?*

**Response:** We thank the Reviewer for this suggestion, however, we find it useful to look at the actual PBLH spatial distribution and seasonal evolution as a measure of the qualitative performance of each product.

*P15L281: Is there actually anywhere in the manuscript some information on what data amounts / number of occultations you compare? And is the Spire NOAA data set the same size / number of occultations as the Spire NASA one? Are they covering the same occultations, sometimes, or always? Somewhere above you mentionoverlapping Spire data, but is that only in time, or are these also the same occultations?*

**Response:** In the revised manuscript, we now include a table (Table 1) which lists the monthly average count of all available RO profiles over the Arctic Ocean for each product. The Spire NOAA dataset has fewer samples compared to Spire NASA, and the daily profile count is dependent on their respective purchase agreements with the vendor. There are some overlapping RO profiles between the two datasets on a given day. In Figure 1(b) of the revised manuscript, a common sub-sample of radio occultations from both Spire NOAA and Spire NASA are compared, which are essentially the same occultations.

*P16L309: The "contemporaneous" is not the correct term here, as the data you looked into are frozen in 2016. The contemporaneous Metop data (NRT) does show verydifferent characteristics.*

**Response:** The word "contemporaneous" is no longer used in the revised manuscript.

*P19L404: This ROM SAF data set contains PBL height estimates from bending angles, refractivity, etc. Have those at all been looked at within this investigation?*

**Response:** No, we did not compare against the PBLH variable within the MetOp data files. Our investigation is focused on the Arctic Ocean, for which a region-specific refractivity-based PBLH retrieval algorithm has been previously developed and tested (Ganeshan and Wu 2015). Hence, we chose to use the established region-specific algorithm to derive PBLH rather than use generic global PBLH data.

**Minor Comments:**

*P4L127: The cited work looking at the 500m cut off height is based on COSMIC-1 only. Here you look at different processing algorithms, different instruments, can youfurther justify that the 500m is applicable here too? ... Actually, reading further, I realized there is justification below. I would recommend to mention this justification herealready, otherwise one is in the blue for a while on why 500m is chosen and mentioned several times.*

**Response:** Thank you to the Reviewer for this comment. We have now included a line in the Data and Methodology section (Line 167 of the revised manuscript), explaining that the sensitivity to cut-off altitude threshold will be explored.
"In this study, sensitivity of commercial RO products to the choice of cut-off altitude threshold will be additionally explored."

*P9L239: Is the Spire data N. Bowler looked at the same as the one you are using, and this statement applies? Spire also updates their processing algorithms.*

**Response:** The Level 2 Spire data in the Bowler (2020) study is also provided by the vendor (Spire), similar to the NASA Spire Level 2 profiles. While we cannot verify the version of the Level 2 product, we expect similarities between the two.

*P15L292: Again, I think statements that are general about Spire in comparison to other missions need to be treated with care. If I got enough money, I could ask Spire to give me 10k occs a day that reach 300m, and they will provide this every day. So no seasonal variability is visible in that data set, as Spire has selected the occultations from a larger set, that might show the seasonal variability. On the other hand, one cannot easily task a Metop satellite to provide more occultations, so that any variability is reduced.*

**Response:** We agree that seasonal variability in the number of RO profiles should only be considered along with the caveat that Spire is contractually obligated to provide a certain number of L2 profiles throughout the year. However, the seasonal variability in the penetration probability (which is a percentage of total profiles reaching a certain altitude) is more likely due to atmospheric conditions. All RO products, including COSMIC-1 and NOAA Spire, show a seasonal reduction in penetration probability during summer months at 500 m altitude and lower. Hoewever, there is no such seasonality in NASA Spire observations which is duly reported in the revised manuscript (Fig. 4).

*Fig 11: Isn't the left plot already shown above? Might be thus better to just show the difference between 300 and 500m cut off.*

**Response:** We no longer show this figure. Instead, the revised manuscript has a similar comparison (Fig. 10) for a different month (Feb 2024) along with a panel showing NOAA Spire derived PBLH for the same month.

**Editorial Comments:**

*P4L114: I assume the reference to this data set should be stated here (it is in the references section, but never used).*

**Response:** This reference has been added. We thank the Reviewer for their attention to detail.

*P17L352: I assume this should be Bowler, N.E.?*
**Response:** This has been corrected.

*P18L367: Is Jarraud cited anywhere?*
**Response:** This reference has been deleted.

*P18L383: Is Maturilli cited somewhere?*

**Response:** This reference has been deleted.

*P19L386: Is Maennel cited somewhere?*
**Response:** This reference has been deleted.

*P19L415: Isn't this the "THE NASA PBL INCUBATION STUDY TEAM REPORT" document, e.g. available here: https://smd-cms.nasa.gov/wp-content/uploads/2023/05/NASA_PBL_Incubation_Final_Report_2.p*

**Response:** Yes. It has been duly noted.

**References:**

Innerkofler, J., G. Kirchengast, M. Schwärz, C. Marquardt, and Y. Andres (2023). GNSS radio occultation excess-phase processing for climate applications including uncertainty estimation, *Atmos. Meas. Tech., 16,* 5217–5247, https://doi.org/10.5194/amt-16-5217-2023, 2023.